# The geography and inter-community configuration of new sexual partnership formation in a rural South African population over fourteen years (2003–2016)

**Hae-Young Kim**[1,2,3]*, **Diego Cuadros**[4], **Eduan Wilkinson**[3], **Dennis M. Junqueira**[3], **Tulio de Oliveira**[3], **Frank Tanser**[2,5,6,7]

1 Department of Population Health, New York University Grossman School of Medicine, New York, NY, United States of America, 2 Africa Health Research Institute, KwaZulu-Natal, Durban, South Africa, 3 KwaZulu-Natal Research Innovation and Sequencing Platform (KRISP), KwaZulu-Natal, Durban, South Africa, 4 Department of Geography and Geographic Information Science, University of Cincinnati, Cincinnati, OH, United States of America, 5 Lincoln International Institute for Rural Health, University of Lincoln, Lincoln, United Kingdom, 6 School of Nursing and Public Health, University of KwaZulu-Natal, Durban, South Africa, 7 Centre for the AIDS Programme of Research in South Africa (CAPRISA), University of KwaZulu-Natal, Durban, South Africa

* hae-young.kim@nyulangone.org

**Data Availability Statement:** The data underlying the results presented in the study are available from the AHRI Data Repository (https://data.

## Abstract

Understanding spatial configuration of sexual network structure is critical for effective use of HIV preventative interventions in a community. However, this has never been described at the population level for any setting in sub-Saharan Africa. We constructed the comprehensive geospatial sexual network among new heterosexual partnerships in rural KwaZulu-Natal, South Africa. In the Africa Health Research Institute (AHRI)'s population-based surveillance, we identified stable sexual partnerships among individuals ($\geq$15 years) from 2003 to 2016. Sexual partnerships and residency were recorded via household surveys (every 4–6 months). We geolocated residents and migration events and mapped the geospatial linkages of sexual partners at the start of sexual partnerships. In a grid composed by 108 cells (nodes; 3kmx3km per cell) covering the surveillance area (438km$^2$), we calculated the degree of connectivity and centrality of the nodes and examined their association with HIV prevalence and incidence per cell. Of 2401 new sexual partnerships, 21% (n = 495) had both partners living within the surveillance area at the start of sexual partnerships, and 76% (376/495) were linked to the geographic HIV cluster with high HIV prevalence identified in a peri-urban community. Overall, 57 nodes had at least one connection to another node. The nodes in the peri-urban cluster had higher connectivity (mean = 19, range: 9–32), compared to outside the cluster (6, range: 1–16). The node's degree of connectivity was positively associated with HIV prevalence of the cell (Pearson correlation coefficient = 0.67; p <0.005). The peri-urban cluster contained nine of the 10 nodes that composed of a single large central module in the community. About 17% of sexual partnerships (n = 421) were formed between a resident and a non-resident partner who out-migrated. Most of these non-resident partners lived in KwaZulu-Natal (86.7%), followed by Gauteng (9.7%), and the median distance between a resident and a non-resident partner was 50.1km (IQR: 23.2–177.2). We

africacentre.ac.za) for researchers who meet the criteria for access to confidential data and sign on the agreement according to the AHRI's protocol for data sharing. The authors confirm they did not have any special access or privileges that others would not have.

**Funding:** This work was supported by three National Institute of Health (NIH) grants (R01HD084233, R01AI124389, and R21TW011687 (Recipient: FCT)). The Africa Health Research Institute's Demographic Surveillance Information System and Population Intervention Programme is funded by the Wellcome Trust (201433/Z/16/Z) and the South Africa Population Research Infrastructure Network (funded by the South African Department of Science and Technology and hosted by the South African Medical Research Council). The funders had no role in study design, data collection and analysis, decision to publish, or preparation of the manuscript.

**Competing interests:** All authors have no conflict of interests to report.

found that the peri-urban HIV cluster served as the highly connected central node of the network for sexual partnership formation. The network was also connected beyond the surveillance area across South Africa. Understanding spatial sexual network can improve the provision of spatially targeted and effective interventions.

## Introduction

In East and Southern Africa, sexual transmission in the general population, especially within stable couples, has been proposed as one of the major drivers of the HIV epidemic [1–4]. A few studies have examined characteristics of sexual networks and transmission, including age-mixing [5, 6], concurrency [7, 8], or multiple partnerships [9]. Some modeling studies have simulated sexual networks and examined the effect of these network characteristics on HIV transmission and prevention [10, 11]. However, understanding and configuring a sexual network in the general population is complex and rarely feasible, and often ignores the spatial dynamics of the sexual network, or "where" partnership formation takes place.

One of the key factors contributing to the dynamics of the sexual contact network is mobility. In sub-Saharan Africa (SSA), individuals have been increasingly mobile away from their homes [12]. For example, in rural KwaZulu-Natal South Africa, more than 20% of household members reported short-term (<10 nights away from the household) or long-term migration within two years, seeking employment or education to nearby cities or adjusting to changes in partnership status [13, 14]. While mobile individuals may maintain their connections to social and sexual networks at origins, they are more likely to experience sexually risky behaviors, risk of HIV acquisition and non-engagement in care at their new destinations [12, 15–17]. Many individuals also in-migrate into rural communities. A study reported that between 20–60% of new HIV infections that occurred in rural KwaZulu-Natal were estimated as external introductions from outside the local area [18]. In the population cohort in Rakai, Uganda, about 30% of HIV-negative participants were considered in-migrants and experienced an increased risk of HIV acquisition [19].

Furthermore, HIV epidemics in SSA are spatially structured and geographically heterogeneous [20–22]. The risk of HIV acquisition among individuals is strongly affected by uneven geospatial and community-level factors, including population viral load and community transmission index [23]. Spatial structure and connectivity of contact networks can substantially drive disease diffusion and transmission in a community. A recent study in Rakai, Uganda, characterized the geography of migration networks and movement of HIV-positive individuals into and out of areas of higher HIV prevalence; however, the study did not directly examine the spatial sexual contact network or connectivity for sexual partnerships [24]. Also, several studies have reported spatial clustering of HIV incidence or prevalence [20, 25, 26] but did not examine whether such clusters may contribute to HIV transmission through a sexual contact network.

Against this background, the longitudinal population-based surveillance at Africa Health Research Institute (AHRI) provides a unique data source to understand the spatial configuration of the sexual network and connectivity in a community suffering a hyper-endemic HIV epidemic and highly mobile. The objective of this study was to quantify and analyze the geospatial configuration and characterization of the sexual contact network emerging from new stable sexual partnership formation in a HIV hyper-endemic rural community in South Africa.

## Methods

### Study setting

We followed up individuals using the population-based longitudinal cohort at AHRI from 2003 to 2016. The study site is located in a rural part of the uMkhanyakude district, KwaZulu-Natal in South Africa, where HIV burden is among the highest in South Africa, with an adult HIV prevalence estimated to be >30% in 2017 [27]. A detailed description of this cohort has been published elsewhere [28]. Briefly, the surveillance is an open cohort of over 100,000 household members from ~12,000 households in the 438 km$^2$ demographic surveillance area. Main reasons for migration out of the surveillance area are formal employment, education, or changes in partnership [29, 30].

The surveillance consists of two components: household and individual surveys. Households are defined as social groups of individuals who largely share the same resources, have one household head, and know the basic information of each other such as name, relationships to the household head, and place of residence [28]. All registered individuals in the study area must be a member of at least one household. During household visits (every 4–6 months), each household head is asked about the residential status and sociodemographic information of household members. Residents keep their day-to-day belongings and normally sleep at the homestead, while non-residents are the household members who moved away from the study area, do not physically reside in the household but maintain social and physical connections via returned visits and/or support at the time of surveys. For non-resident members, information on their destinations for migration was recorded from other household members during household surveillance. All resident members aged ≥15 years are eligible to receive annual HIV testing after written informed consent are obtained. After obtaining written informed consent for HIV testing, field workers collected finger-prick blood samples and prepared dried blood spots according to the WHO and UNAIDS *Guidelines for Using HIV Testing Technologies in Surveillance* [31]. Written informed consent was also obtained for participation in the surveillance including all study procedures, data collection, and analysis from all individuals ≥ 18 years, whereas parental/guardian consent with child assent was sought for eligible individuals aged 15 to 17 years.

### Participants and procedures

During household surveillance, detailed information on stable sexual partnerships, including the start and end dates of the relationships, was separately sought from all female household members aged ≥ 15 years. Participation in the household-level surveillance, including data collection on sexual partnership information, was extremely high (>98%) and remained stable over time [32]. We defined the relationship as being "stable" if the partners in the relationships belonged to the same household (i.e., conjugal relationships) either by forming a new household or becoming a household member of one partner's exiting household. These relationships include both marital and non-marital relationships, tend to last longer, and are socially recognized as couples while casual partnerships are transient and do not become the same household members. Information on male partners registered in the same households was therefore also obtained during the household surveillance and linked using their individual identifiers, and a unique identifier was given to each stable sexual partnership. Each individual included in the surveillance is geo-located to the person's homestead of residence using the comprehensive geographic information system (with accuracy < 2m) [33]. Although individuals may belong to multiple households simultaneously, they can only be recorded as living in one homestead at any given time. We included all female and male partners in stable heterosexual partnerships newly formed between 2003 and 2016.

Naturally, couples may take time to decide to belong to the same household or cohabitate after initiating a sexual partnership. Thus, we followed up and further examined the time to household formation and residence in the same homestead (i.e., cohabitation) since the start of their sexual relationships. Same household members do not necessarily live in the same homestead. If a partner moved into an existing homestead, we assumed the partner's moving-in date to that homestead as the initiation of cohabitation.

We categorized participants into three types of migration based on their geospatial locations and status of household membership at the start of stable sexual partnerships: 1) internal migrant if the individual was already registered at least under one household and living within the study area; 2) external in-migrant if the individual was not registered nor lived in the study area but later moved into a homestead in the study area and registered as a household member after the stable partnership formation; 3) out-migrant if the individual had been registered as a household member but was a non-resident or had no prior history of residency in the study area.

## Spatial network configuration

We generated a grid composed of 108 cells of 3km X 3km dimension, which covered the entire surveillance area (S1 Fig; Fig 1). Using the geographical location of individuals at the start of partnership formation, we conducted several network analyses to identify network structure characteristics. First, we assessed the association between the node degree (i.e., the number of connections that a node has to other nodes in the network–the higher the degree is, the more connected the node is) of each cell and the HIV prevalence estimated for the corresponding cell using Pearson correlation coefficient and bivariate maps (Fig 2). Second, we calculated the eigenvector centrality of the node, which is a measure of its influence in the network based on the degree of being linked to other highly connected nodes (i.e., the higher eigenvector score is, the more connected the node is to other nodes which also have high eigenvector scores themselves). Eigenvector centrality was normalized in the range of 0 to 1. Lastly, we conducted a community structure detection analysis to identify modules (communities) of nodes densely connected among themselves but sparsely connected to other modules of nodes (Fig 3).

## Statistical analysis

A geographical cluster with a high burden of HIV incidence and prevalence within the study area was previously identified through Kulldorff spatial scan statistics analysis [34] using the 2011–14 population-based HIV surveillance dataset [34, 35]. The methodology has been widely used in various public health research [36–39] and described in detail elsewhere [36]. Briefly, this technique is based on a cluster detection test designed to identify areas with higher numbers of cases (i.e., HIV-positive individuals) than expected under the assumption of random distribution of the cases in space, controlling for the uneven underlying population density in the study area. The analysis detects potential clusters by gradually scanning a circular window that scans across the entire study region. Then, the statistical significance of each potential cluster was estimated using a likelihood ratio test. Geographical clusters with a *p-value* < 0.05, calculated through Monte Carlo simulations, were identified as statistically significant. The cluster occurred in a dominantly peri-urban community adjacent to the National Road [25, 40].

We compared the proportion of different types of migration in newly formed stable sexual partnerships among female and male partners, and p-values are reported using a chi-square test of independence [41]. The time trend of migration types was examined between 2003 and 2016 by gender (S2 Fig). For the stable sexual partnerships in which partners were both

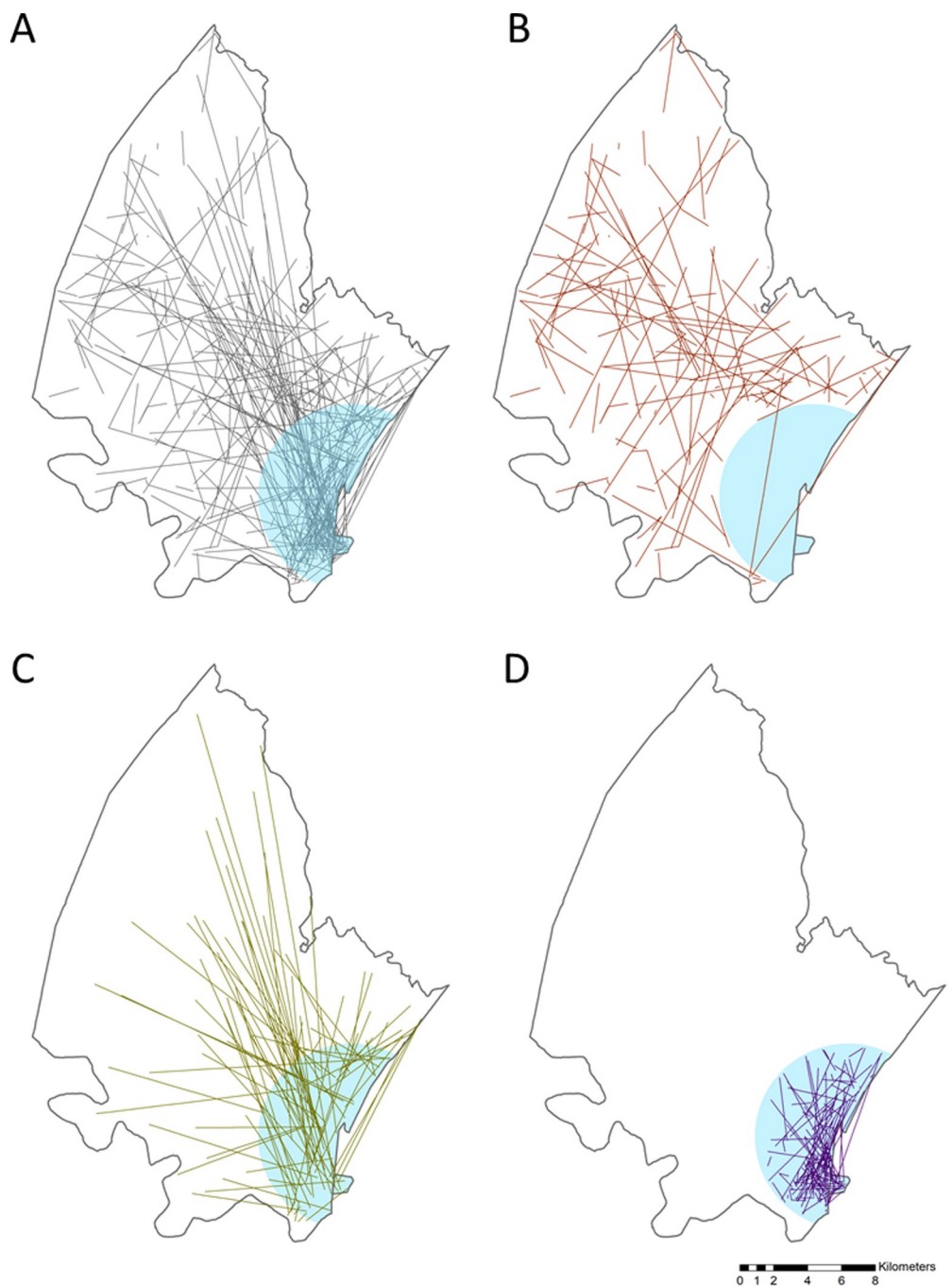

**Fig 1. Linkage of female and male partners of stable sexual relationship pairs in which both partners were residents at the start of sexual partnership formation.** A) All links formed by partners in which both where located within the surveillance area at the beginning of the relationship (n = 495); B) formation links where both partners were located outside the peri-urban HIV cluster area (n = 117); C) formation links where only one partner is located within the peri-urban HIV cluster area (n = 148); D) formation links where both partners were located inside the peri-urban cluster area (n = 230). The location of the peri-urban HIV cluster area is represented by the blue area. Only approximate locations are shown in the map to protect confidentiality of respondents.

residents within the study area, each partner's homestead geographical location was linked to the other partner's location at the start of sexual partnership. The distance between the linked geolocations of homesteads was calculated (Fig 1). We also calculated the proportion of stable

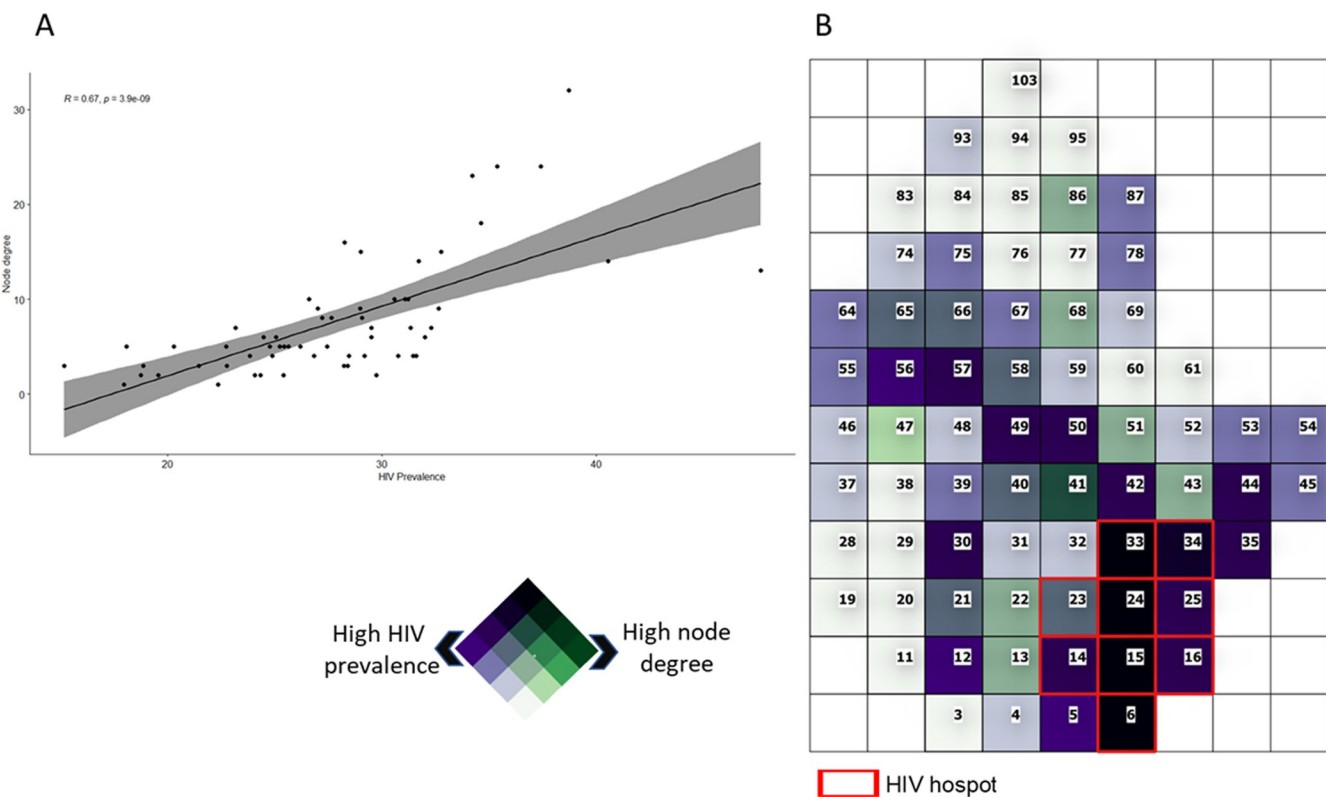

**Fig 2. Association between node degree and HIV prevalence of the corresponding cell.** A) Correlation between HIV prevalence of the node and the corresponding node degree. The line illustrates the regression line showing the positive association between the node connectivity (degree) and the HIV prevalence of the corresponding cell (p<0.005); B) Bivariate map showing areas with high node degree and high HIV prevalence (dark cells) located withing the peri-urban cluster delineated by red lines. Numbers within the cells illustrate the labels of each cell (node). The nodes included within the HIV cluster are delineated in red.

sexual partnerships within the overlaid peri-urban cluster with the highest HIV burden in the study area. For privacy protection and ethical considerations, we generated maps without the inclusion of any geographical reference. The maps are shown for illustrative purposes only. All analyses were conducted using STATA 15.0 [42] and R. Spatial maps were generated using ArcGIS 10.7.1 [43].

## Ethical approval

Funding sources had no role in the decision to prepare the manuscript and submit it for publication. The permission for the demographic and HIV surveillance was approved from the Biomedical Research Ethics Committee at University of KwaZulu-Natal (BE290/16).

## Results

Since January 2003, 2,401 new stable sexual partnerships were formed, linking 2,401 females and 2,295 males. At the start of sexual partnership formation, the median age of females and males was 25 (interquartile range [IQR]: 20, 31) and 31 (IQR: 25, 39), respectively. Of 2,401 sexual partnerships, each of which progressed to belonging to the same household, 2,195 (91.4%) had one household membership, while 209 (8.7%) had more than one household membership over time. A majority (n = 2,214 couples) formed a sexual partnership first then became members of the same household later. The median time between the start of sexual

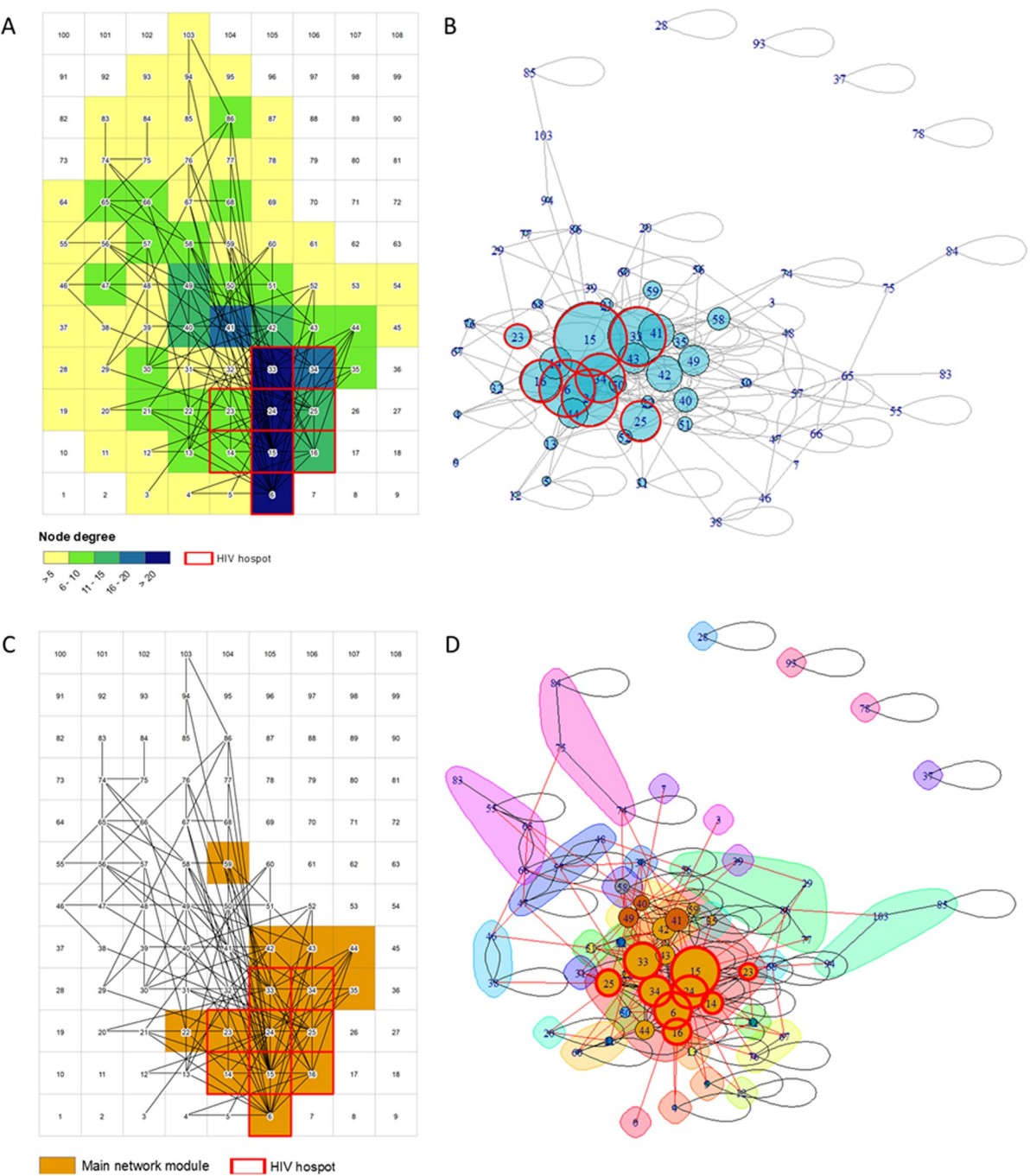

**Fig 3. Spatial network configuration of stable sexual partnerships.** A) Map illustrating the node degree of each pixel. Nodes with high degree are illustrated in blue, whereas nodes with low degree are illustrated in yellow; B) Eigenvalue estimations for each node. Nodes with highest eigenvalues are centered close to the nodes with their strongest connections. Circles delineated in red indicate the nodes located within the HIV cluster; C) Map illustrating the location of the cells included in the main node community of the network (brown cells); D) Illustrates the different node communities identified in the network. Circles delineated by red illustrate the nodes located within the peri-urban HIV cluster. The numbers in all figures represent the labels of each node in the network.

partnership and recorded household membership was 2.0 (IQR: 0.9, 4.0) years. Despite belonging to the same household, 441 (18.2%) pairs reported having never lived in the same homestead under the reported sexual partnerships. Of 1,961 (81.7%) cohabitated as sexual

**Table 1. Types of migration in sexual partnership formation by gender.**

| Number (%) | Male | | | |
|---|---|---|---|---|
| Female | Internal migrant | External in-migrant | Out-migrant | Total |
| Internal migrant | 495 | 108 | 240 | 843 (35%) |
| External in-migrant | 359 | 215 | 301 | 875 (36%) |
| Out-migrant | 296 | 94 | 293 | 683 (29%) |
| Total | 1150 (48%) | 417 (17%) | 834 (35%) | 2401 |

partners, the median time between the start of sexual partnership and cohabitation was 2.0 (IQR: 0.9, 4.0) years. Overall, about 74% (1773/2401) started cohabitation within six months after establishing the same household membership, 2.7% within 6–12 months, and 5.2% after one year.

When we examined the geolocations of new sexual partners at the start of sexual partnership formation, 21% (495/2,401) of both partners were internal migrants (Table 1). The percentage of being an internal migrant was higher among males (48%) than among females (35%) (p<0.01). Of females, 36% (875/2,401) were external in-migrants while 28% (638/2,401) were out-migrants. Of males, 17% (417/2,401) were external in-migrants, and 35% (834/2,401) were out-migrants. There was a general trend that internal migration increased for males, but not for females between 2003 and 2016 (S2 Fig).

When we geographically mapped those who internally moved within the surveillance area, we found that out of the 495 stable partnerships generated inside the surveillance area, 76% (n = 378) had at least one partner living within the peri-urban cluster when initiating the sexual partnerships (Fig 1). Of these, 230 stable sexual partnerships had both partners living within the HIV cluster, with a mean distance of 1.3 km between partners. The mean distance was 9.1 km among the 148 stable sexual partnerships that had only one partner living within the cluster. The average distance was 3.4 km among stable sexual partnerships whose partners were both living outside the peri-urban cluster.

Network configuration resulted from the spatial grid aggregation is illustrated in S1 Fig. Excluding the 10 nodes which had no connections, node degree ranged from one to 32. The average degree of the nodes located within the peri-urban HIV cluster was 19 (range: 9–32), whereas the average degree of nodes located outside the peri-urban HIV cluster was 6 (range: 1–16) (Fig 2A). We found a statistically significant positive correlation between the node degree and the average HIV prevalence of the cell (Pearson correlation coefficient = 0.67; p < 0.005; Fig 2A). The bivariate map indicated a high concentration of cells with high HIV prevalence and high node degree within the peri-urban HIV cluster (Fig 2B). Nodes 6, 15, 24, 33, and 34, which were located within the peri-urban HIV cluster, had the highest HIV prevalence ranging from 34.2% in node 33 to 38.7% in node 15, and the high node degree ranging from 18 links in node 34 to 32 links in node 15.

Eigencentrality analysis showed that node 15 had the highest eigenvector centrality (1.00), followed by node 24 (0.79), node 6 (0,73), node 33 (0.72), and node 34 (0.68). All of them were located within the peri-urban HIV cluster (Fig 3B). This network characteristic generated a central community composed by nodes that most of them were located within the peri-urban cluster (Fig 3C). Community detection analysis identified one single large central module formed by nodes 6, 14, 15, 23, 24, 25, 33, 34, 42, and 44 (Fig 3D). Nine of these 10 nodes were located within the HIV high-risk location. The other nodes in the network were sparsely connected, and only connected through the large central module identified. For example, a small network module formed by nodes 84, 74, and 75 was not directly connected with another

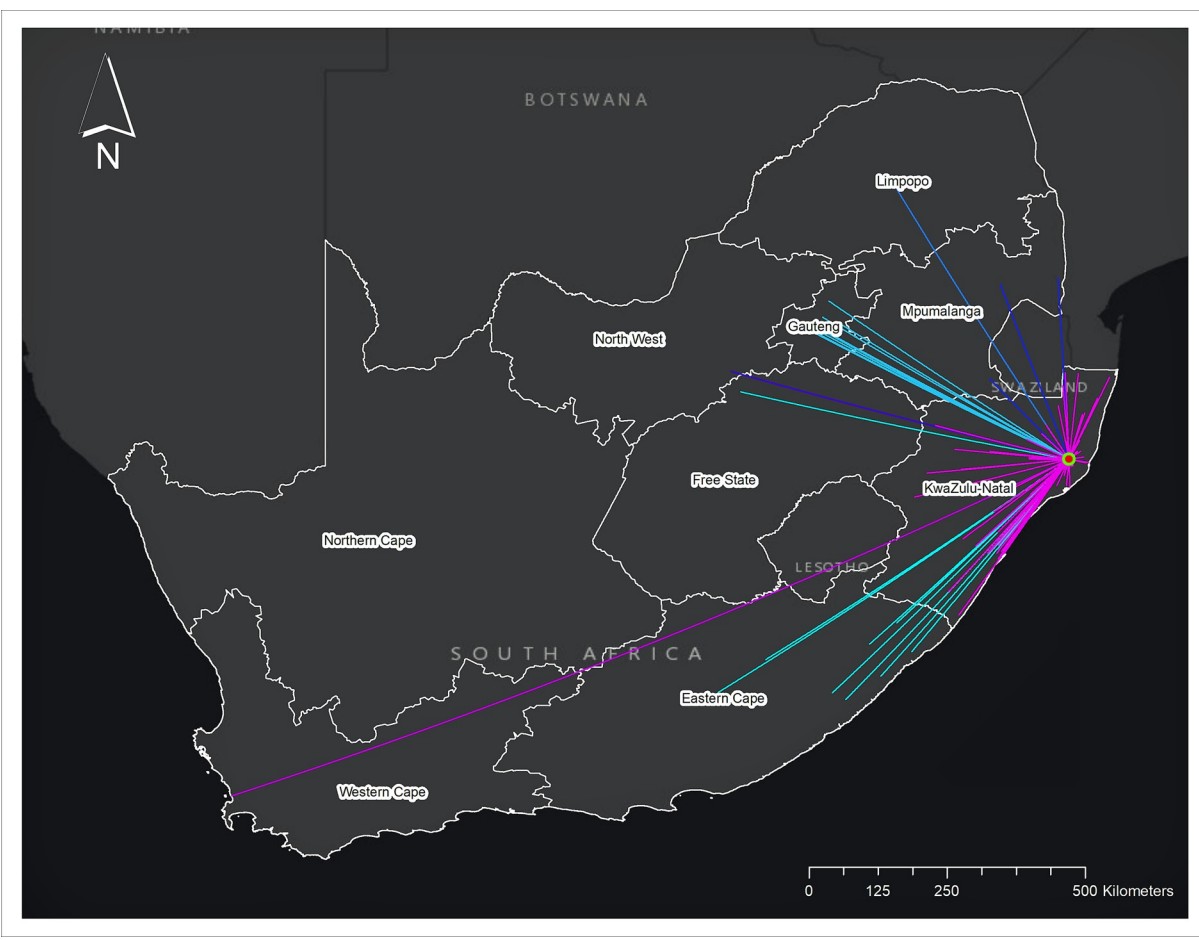

**Fig 4. Linkage of female and male partners of stable sexual relationship pairs between a resident and a non-resident of the surveillance area.** All links were formed by relationship pairs in which one partner was a resident within the surveillance area and the other partner was a non-resident outside the surveillance area across South Africa (n = 421). Linkage to a non-resident is represented by different colors of the states. Only approximate locations are shown in the map to protect confidentiality of respondents.

small module formed by nodes 85, 94 and 103, but they were indirectly connected through the large central module located within the peri-urban cluster.

When we mapped the geospatial linkages of partnerships in which one partner was a non-resident at the start of sexual partnership formation (Fig 4), 85.7% of non-residents in the partnerships were in KwaZulu-Natal, followed by 9.7% in Gauteng and 2.6% in Eastern Cape. The median distance between a non-resident partner and a resident partner was 50.1 km (IQR: 23.3,177.2).

## Discussion

To our knowledge, this is the first study to reconstruct the spatial structure of the sexual contact network formed by heterosexual partnerships in a highly mobile and HIV hyper-endemic setting. First, we found surprisingly high connectivity (76%) of the stable sexual partnerships to the peri-urban HIV cluster in the study area. This result is remarkably similar to the finding from the HIV transmission network constructed using phylogenetic analysis that 70% of the HIV transmission links were directly linked to the peri-urban HIV cluster [35]. Also, a central module of the network was located in the peri-urban HIV cluster, indicating substantial

mixing between the population in the peri-urban cluster and the entire community. Second, about 80% of stable sexual partnerships had at least one partner as an external in-migrant or a non-resident at the start of sexual partnership. The network between out-migrants and residents was spread across the country, reaching as far as Western Cape. Collectively, these results suggest that the central network module in the peri-urban cluster might behave as the highly connected hub of the sexual contact network and play a key role for the HIV transmission in the entire study area and beyond.

These findings have several important policy implications. First, we previously reported that although the peri-urban cluster bounded only about 8% of the entire surveillance area and enclosed 30% of the total population, more than 70% of the HIV transmissions were phylogenetically linked to the peri-urban cluster [35]. This peri-urban cluster is characterized by enhanced economic activities surrounding commercial areas and proximity to the national highway. In this study, we recorded a similar pattern, in which more than half of new stable sexual partnerships formed among residents within the surveillance area were linked to this HIV cluster. We also found that approximately 60% of the external in-migrants settled down in this geographical area when they first moved into the study area (S3 Fig). The high connectivity of this peri-urban cluster can facilitate the diffusion of the virus across the entire contact network through the partnership formation among both internal members and external in-migrants. Several studies have proposed targeting resources and interventions to geographically clustered areas with high HIV prevalence and incidence [26, 44, 45] or populations at high risk of HIV infection [46]. Our study findings suggest that improving access to HIV prevention, testing, and care in this geographically clustered area could not only provide health benefits at the individual-level but also be an effective way to prevent HIV transmission by disrupting the connectivity of the entire contact network in the community and maximize resources needed to achieve epidemic control. So far, limited studies have tested the effectiveness of geographically targeted programs. Further studies are needed to develop and test the feasibility of geographically targeted intervention programs and their effectiveness in reducing HIV transmission and epidemic.

Second, the current study shows that about 80% of the sexual partnerships had at least one partner who was an external member or an out-migrant, most of whom relocated to the study area after initiating the sexual partnership. This suggests that many individuals likely meet their partners while they are away for a short or long-term period then would move back into the community with their sexual partners. A recent study established that up to 60% of new HIV transmissions in the study area were attributed to external introduction using phylogenetic analysis [18]. There is limited data on migration driven by long-term relationships or marriage in SSA but a few studies have shown that women are more likely to relocate to their partners' places or due to changes in marital status [47, 48]. External in-migrants who are new to the local community might have an increased risk of HIV acquisition and experience lack of access to care [19]. However, little is known about how external in-migrants in the context of stable sexual partnerships are affected by HIV epidemics or navigates access to health care [49]. Further studies need to investigate the best ways to engage new migrants into HIV prevention and treatment services in order to optimize intervention efforts among them and the wider community.

It is also noteworthy that over 30% of stable sexual partners at the start of the partnerships were non-residents of the study area, majority of them were living elsewhere in KwaZulu-Natal or Gauteng. The phenomenon of prevalent circular migration in South Africa has been well described [47, 50]. In the study area, more than 20% of population experience short-term and long-term mobility within two years, and individuals have been increasingly mobile in recent years, seeking economic or education opportunities [13, 14]. Mobile population away

from the community are at a higher risk of HIV acquisition and risky sexual behaviours [15] and less likely to engage in care [12]. Such a high proportion of circular migrant would also facilitate sexual transmission network across the country beyond this community.

Lastly, we observed about average two years between the initiation of sexual partnership and household formation and/or cohabitation, and the couples may be living far apart outside the community. This implies that couples-based interventions for HIV preventive services at home or clinics might be less effective in the initial years of sexual partnerships and need to incorporate such geographical barriers. Further, even after forming a household, nearly 20% of the stable sexual partners never reported having the same residency during the entire sexual partnership. Novel approaches to effectively engage partners who are mobile and physically separated would be warranted.

The study has some limitations worth noting. Since we did not collect demographic information for casual sexual partners, we would have missed network patterns of some casual partnerships including transactional sex. The median age of partners in this analysis was significantly higher than the average age of first-time sexual partners in the surveillance. However, we presume that most of these stable sexual partnerships initiated as casual partnerships, which later became more stable and long-term relationships, therefore representing both casual and long-term sexual partnerships in terms of geography and sexual contact networks. Also, the start and end dates of sexual partnerships was self-reported thus could have been inaccurate. However, as most partners were living together, we believe under- or over-reporting of actual partnership formation would have been limited. Lastly, we only assessed the spatial configuration at the time of sexual partnership formation. Future studies can examine the spatial dynamics of partnership formation and dissolution over a longitudinal period.

## Conclusions

We found that the peri-urban HIV cluster served as the highly connected central community of the network for sexual partnership formation. More than half of new stable sexual partnerships were geographically linked into the peri-urban HIV cluster for both internal and external in-migrants. Understanding such spatial configuration of sexual network can improve the provision of effective interventions.

## Supporting information

**S1 Fig.** Links generated by the formation of the stable sexual partnerships among individuals located within the surveillance area (A), where aggregated in nodes formed by the centroids of the pixels generated by a 3km x 3km grid that covered the entire surveillance area (B). Cells delineated in red illustrate are located within the peri-urban cluster area (red area in A). Nodes were labeled using sequential numbers.
(TIF)

**S2 Fig.** Types of migration in sexual partnership formation by gender in 2003–2015: (A) Females and (B) Males. Blue circle, red diamond, and green cross symbols represent internal migrants, external in-migrants, and out-migrants, respectively.
(TIF)

**S3 Fig.** Density of in-migration into which external migrants from outside surveillance area moved by gender: (A) Females and (B) Males. Continuous surface maps of the density of external in-migrants were generated using a moving two-dimensional Gaussian kernel of 3 km search radius to produce robust density estimates that vary across continuous geographical space to generate a grid of 100m x 100m pixels. The size of the kernel was determined from the

results of previous work [33]. The kernel moves systematically across the map and measures spatial variation in the density of in-migrants across the surveillance area.
(TIF)

## Acknowledgments

We would like to thank all study participants and study staff for their time and dedication.

## Author Contributions

**Conceptualization:** Hae-Young Kim, Diego Cuadros, Frank Tanser.

**Data curation:** Hae-Young Kim, Eduan Wilkinson, Dennis M. Junqueira, Tulio de Oliveira, Frank Tanser.

**Formal analysis:** Hae-Young Kim, Diego Cuadros, Eduan Wilkinson, Dennis M. Junqueira.

**Funding acquisition:** Frank Tanser.

**Investigation:** Hae-Young Kim, Diego Cuadros, Frank Tanser.

**Methodology:** Hae-Young Kim, Diego Cuadros, Eduan Wilkinson, Dennis M. Junqueira, Tulio de Oliveira, Frank Tanser.

**Software:** Hae-Young Kim.

**Supervision:** Tulio de Oliveira, Frank Tanser.

**Validation:** Hae-Young Kim, Diego Cuadros, Frank Tanser.

**Visualization:** Hae-Young Kim, Diego Cuadros.

**Writing – original draft:** Hae-Young Kim, Diego Cuadros.

**Writing – review & editing:** Hae-Young Kim, Diego Cuadros, Eduan Wilkinson, Dennis M. Junqueira, Tulio de Oliveira, Frank Tanser.

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
