## [Decision Letter · Decision Letter 0]

23 Sep 2021

 PGPH-D-21-00186 The geography and inter-community configuration of new sexual partnership formation in a rural South African population over fourteen years (2003-2016) PLOS Global Public Health

Dear Dr. Kim,

Thank you for submitting your manuscript to PLOS Global Public Health. After careful consideration, we feel that it has merit but does not fully meet PLOS Global Public Health’s publication criteria as it currently stands. Therefore, we invite you to submit a revised version of the manuscript that addresses the points raised during the review process.

We look forward to receiving your revised manuscript.

Kind regards,

Bernard Cazelles, Ph.D.

Academic Editor

Journal Requirements:

1. You indicated that you had ethical approval for your study. We note that written informed consent was obtained for HIV testing. In your Methods section, please state whether consent was obtained for participation in this study, including all procedures and data collection and analysis.

In addition, please ensure you have also stated whether you obtained consent from parents or guardians of the minors included in the study or whether the research ethics committee or IRB specifically waived the need for their consent.

2. We ask that a manuscript source file is provided at Revision. Please upload your manuscript file as a .doc, .docx, .rtf or .tex. If you are providing a .tex file, please upload it under the item type ‘LaTeX Source File’ and leave your .pdf version as the item type ‘Manuscript’.

3. Please amend your detailed Financial Disclosure statement. This is published with the article, therefore should be completed in full sentences and contain the exact wording you wish to be published.

i). Please include all sources of funding (financial or material support) for your study. List the grants (with grant number) or organizations (with url) that supported your study, including funding received from your institution. 

ii). State the initials, alongside each funding source, of each author to receive each grant.

iii). State what role the funders took in the study. If the funders had no role in your study, please state: “The funders had no role in study design, data collection and analysis, decision to publish, or preparation of the manuscript.”

iv). If any authors received a salary from any of your funders, please state which authors and which funders.

4. Please ensure that the funders and grant numbers match between the Financial Disclosure field and the Funding Information tab in your submission form. Note that the funders must be provided in the same order in both places as well.

5. Please provide separate figure files in .tif or .eps format only, and remove any figures embedded in your manuscript file. If you are using LaTeX, you do not need to remove embedded figures.

For more information about figure files please see our guidelines: https://journals.plos.org/globalpublichealth/s/figures

6. We notice that your supplementary figures are included in the manuscript file. Please remove them and upload them  with the file type 'Supporting Information'. Please ensure that all Supporting Information files are included correctly and that each one has a legend listed in the manuscript after the references list. 

7. Please provide us with a direct link to the base layer of the map used in Figures 1, 4, Supplementary Figures 1, 3 and ensure this location is also included in the figure legend. 

Please note that, because all PLOS articles are published under a CC BY license (creativecommons.org/licenses/by/4.0/), we cannot publish proprietary maps such as Google Maps, Mapquest or other copyrighted maps. If your map was obtained from a copyrighted source please amend the figure so that the base map used is from an openly available source.

Please note that only the following CC BY licences are compatible with PLOS licence: CC BY 4.0, CC BY 2.0  and CC BY 3.0, meanwhile such licences as CC BY-ND 3.0 and others are not compatible due to additional restrictions. If you are unsure whether you can use a map or not, please do reach out and we will be able to help you. 

The following websites are good examples of where you can source open access or public domain maps:

Additional Editor Comments (if provided):

In light of the reviews, we would like to invite the resubmission of a significantly-revised version that takes into account the reviewers' comments. Another weakness of your manuscript is about statistical approaches and clustering method. More explanation or adapted references are needed.

Reviewers' comments:

Reviewer's Responses to Questions

**Comments to the Author**

1. Does this manuscript meet PLOS Global Public Health’s publication criteria? Is the manuscript technically sound, and do the data support the conclusions? The manuscript must describe methodologically and ethically rigorous research with conclusions that are appropriately drawn based on the data presented.

Reviewer #1: Yes

Reviewer #2: Yes

2. Has the statistical analysis been performed appropriately and rigorously?

Reviewer #1: Yes

Reviewer #2: I don't know

3. Have the authors made all data underlying the findings in their manuscript fully available (please refer to the Data Availability Statement at the start of the manuscript PDF file)?

Reviewer #1: Yes

Reviewer #2: Yes

4. Is the manuscript presented in an intelligible fashion and written in standard English?

Reviewer #1: Yes

Reviewer #2: Yes

5. Review Comments to the Author

Reviewer #1: The paper presents important data on the geographic network structure of stable sexual relationships in rural KwaZulu-Natal, South Africa. While the methods, analysis, and conclusions are reasonable, I think readers would benefit from a more in-depth engagement with previous work on geospatial/network interventions and the implications these current findings have on this work. The integration of sexual network and geospatial data remains relatively limited but geographically targeted interventions have been discussed for many years. Given the high quality data used in the current manuscript, I think it would be beneficial to move beyond vague references to how these data support the notion of targeted interventions and try to propose ways in which this data could be specifically used to inform these efforts. For example, what do the data say about what such interventions should look like or what data would be needed to better inform the development of these interventions? A more thorough discussion of these topics would increase the likelihood of this manuscript leading to development and testing of the interventions proposed in the manuscript.

Reviewer #2: The authors present work looking at the spatial distribution of male-female partnerships in a demographic surveillance site in rural South Africa. Since the site is population-based, this represents a rare opportunity to understand relationship formation in relation to areas of high or low HIV prevalence.

While the approach appears sound and the conclusions are well justified from the results, there's a few areas where greater clarity could help to avoid confusion for the reader.

Major comments

1. p5, relationship information: I was wondering if the authors could provide more details on how the relationship information is collected. Was this from the individual survey of resident women in the household? If that's the case, what does the response rate look like over time? Is it the case that most women can be found for interview, and that most of them consent to the interview? Of those who do consent, do most also respond about their partnerships or are there refusals? This to me is crucial, as a key strength of the study is the population-based aspect, so it would strengthen the paper to provide greater details on the coverage of the partnership data.

2. p5, relationship information: It wasn't clear to me how the male partners were linked through surveillance? Was it through a matching approach or algorithm? More detail on this would also strengthen our understanding of how the relationship data was formed.

3. p5, relationship information: I didn't quite follow how time was handled in the network formation. I see how time is used for other aspects of the analysis, but I didn't quite follow how the network was built with the time aspect. I do note that the authors discuss the limitation of not examining this in a longitudinal way, so I'm more just unsure how the authors are considering time periods in terms of the network at time of partnership formation.

4. p5, p6 network configuration: I was wondering for the two network measures (node degree and eigenvector centrality) if the authors could provide a bit more guidance for the reader on the interpretation? For readers not familiar with these measures, perhaps a small example network where the authors present these two measures and what they mean would make it easier for the reader to then interpret the main results.

Minor comments

5. p7, line 201: The authors note nodes with no connections were excluded, I was wondering if they could report how many there were that were excluded?

6. PLOS authors have the option to publish the peer review history of their article (what does this mean?). If published, this will include your full peer review and any attached files.

**Do you want your identity to be public for this peer review?** For information about this choice, including consent withdrawal, please see our Privacy Policy.

Reviewer #1: No

Reviewer #2: No

---

## [Decision Letter · Decision Letter 1]

22 Nov 2021

The geography and inter-community configuration of new sexual partnership formation in a rural South African population over fourteen years (2003-2016)

PGPH-D-21-00186R1

Dear Dr. Kim,

We're pleased to inform you that your manuscript has been judged scientifically suitable for publication and will be formally accepted for publication once it meets all outstanding technical requirements.

Within one week, you'll receive an e-mail detailing the required amendments. When these have been addressed, you'll receive a formal acceptance letter and your manuscript will be scheduled for publication.

An invoice for payment will follow shortly after the formal acceptance. To ensure an efficient process, please log into Editorial Manager at https://www.editorialmanager.com/pgph/ click the 'Update My Information' link at the top of the page, and double check that your user information is up-to-date. If you have any billing related questions, please contact our Author Billing department directly at authorbilling@plos.org.

Kind regards,

Bernard Cazelles, Ph.D.

Academic Editor

Additional Editor Comments (optional):

Reviewers' comments:

Reviewer's Responses to Questions

**Comments to the Author**

1. If the authors have adequately addressed your comments raised in a previous round of review and you feel that this manuscript is now acceptable for publication, you may indicate that here to bypass the “Comments to the Author” section, enter your conflict of interest statement in the “Confidential to Editor” section, and submit your "Accept" recommendation.

Reviewer #2: All comments have been addressed

2. Does this manuscript meet PLOS Global Public Health’s publication criteria? Is the manuscript technically sound, and do the data support the conclusions? The manuscript must describe methodologically and ethically rigorous research with conclusions that are appropriately drawn based on the data presented.

Reviewer #2: (No Response)

3. Has the statistical analysis been performed appropriately and rigorously?

Reviewer #2: (No Response)

4. Have the authors made all data underlying the findings in their manuscript fully available (please refer to the Data Availability Statement at the start of the manuscript PDF file)?

Reviewer #2: (No Response)

5. Is the manuscript presented in an intelligible fashion and written in standard English?

Reviewer #2: (No Response)

6. Review Comments to the Author

Reviewer #2: (No Response)

7. PLOS authors have the option to publish the peer review history of their article (what does this mean?). If published, this will include your full peer review and any attached files.

**Do you want your identity to be public for this peer review?** For information about this choice, including consent withdrawal, please see our Privacy Policy.

Reviewer #2: No
